# Multidisciplinary Approach for Evaluating the Geochemical Degradation of Building Stone Related to Pollution Sources in the Historical Center of Naples (Italy)

**Valeria Comite** [1,†], **Michela Ricca** [2,†] , **Silvestro Antonio Ruffolo** [2], **Sossio Fabio Graziano** [3], **Natalia Rovella** [2], **Concetta Rispoli** [4] , **Chiara Gallo** [5], **Luciana Randazzo** [2] , **Donatella Barca** [2] , **Piergiulio Cappelletti** [4] and **Mauro Francesco La Russa** [2,*]

[1] Department of Chemistry, University of Milan, 20133 Milan, Italy; valeria.comite@unimi.it
[2] Department of Biology, Ecology and Earth Sciences (DiBEST), University of Calabria, 87036 Arcavacata di Rende, CS, Italy; michela.ricca@unical.it (M.R.); silvestro.ruffolo@unical.it (S.A.R.); natalia.rovella@unical.it (N.R.); luciana.randazzo@unical.it (L.R.); donatella.barca@unical.it (D.B.)
[3] Department of Pharmacy, University of Naples Federico II, 80138 Napoli, Italy; sossiofabio.graziano@unina.it
[4] Department of Earth Science, Environment and Resources (DiSTAR), University of Naples Federico II, 80138 Napoli, Italy; concetta.rispoli@unina.it (C.R.); piergiulio.cappelletti@unina.it (P.C.)
[5] Department of Chemistry and Biology, University of Salerno, 84084 Fisciano, SA, Italy; chgallo@unisa.it
*  Correspondence: mlarussa@unical.it
†  These authors contributed equally to the work.

**Abstract:** Natural stones have represented one of the main building materials since ancient times. In recent decades, a worsening in degradation phenomena related mostly to environmental pollution was observed, threatening their conservation. The present work is focused on the minero-petrographic and geochemical characterization of black crust (BC) samples taken from the historical center of Naples, after selecting two pilot monumental areas. The latter were chosen based on their historical importance, type of material, state of preservation and position in the urban context (i.e., high vehicular traffic area, limited traffic area, industrial area, etc.). The building materials used and their interaction with environmental pollutions were studied comparing the results obtained by means of different analytical techniques such as polarized light Optical Microscopy (OM), scanning electron microscopy with energy dispersion system (SEM-EDS), X-ray powder diffraction (XRPD) and laser ablation coupled with inductive plasma mass spectrometry (LA-ICP-MS).

**Keywords:** black crusts; cultural heritage; marble; Naples pollution; heavy metals

## 1. Introduction

Air pollution strongly affects the integrity of stone materials, since it promotes their degradation over time, especially in an urban context [1–9]. The formation of black crusts, which occurs mainly on carbonate rocks, represents one of the most dangerous degradation forms caused by air pollution [10–18]. Generally, their formation is due to calcium carbonate ($CaCO_3$) sulphation, as a consequence of pH value decrease caused by $SO_2$ in the polluted atmosphere [19]. This dissolution allows gypsum ($CaSO_4 \cdot 2H_2O$) to precipitate, which because of its low water solubility of 2.4 g/L at 25 °C [20] remains as a crust that becomes black (due to soot particles) on surfaces protected from intense wash-out [21–24].

The formation of gypsum on the stone surface is a rapid process and can also be accelerated by the deposition of particulate matter rich in metals and metal oxides, which can act as a catalyst in the sulphation reaction.

Moreover, the black crust shows some differences in microstructure and porosity from the substrate, leading to detachment of the black crust itself and the gradual weakening of the stone of the monument surface. [25–27]. Additionally, carbonaceous particles emitted by combustion processes (dark grey-black color) are the main factors responsible for the blackening of buildings [28].

Recently, the study of black crusts has made some interesting developments [12,27,29–33]; similarly, research related to atmospheric deposit composition has made it possible to understand the major causes of pollution, especially around monuments [34,35].

The characterization of black crusts on built heritage has a dual purpose. On one hand, there is a chance to understand the degree of decay of stone material in terms of microstructural features and chemical and mineralogical compositions. This can be useful for choosing appropriate restoration procedures. On the other hand, these analyses can provide information regarding air pollution in the nearby environment, since the black crusts themselves can act as passive samplers of pollutants, with particular reference to metals [23,24,26–33].

In this paper, the analysis of black crusts from two monumental and historical sites located in the city of Naples is reported. To the best of the authors' knowledge, this is the first time that black crusts have been studied in this city. Naples is located in southern Italy, in the Campania region, and is the third-largest city in Italy following Rome and Milan, with high population density and the environmental consequences related to this. High pollution rates due to intense and slowed motor vehicle traffic characterize the urban downtown. According to the City Council, around 2,348,208 vehicles pass through the urban area every day (data ISTAT 2020). Moreover, the city center is only roughly 10 km away from several industrial areas and is close to the port, one of the most important for passenger traffic and docking of international cruise ships. The historical city is the largest in Europe, and has been designated a UNESCO World Heritage Site [36].

The sites selected for investigations are within the built Heritage of the urban center, namely the complex of *San Domenico Maggiore* and some sculptures of the cloister of *San Marcellino e Festo*, which form part of the religious complex of the same name [37].

The choice of the two sites with different exposures in the urban context of Naples was made in order to detect variability in the black crusts, mainly due to air pollution phenomena. In this regard, the complex of *San Domenico Maggiore* is located in an area with a high degree of vehicular traffic; conversely, the sculptures of the cloister of *San Marcellino e Festo*, although outdoors, are currently in an area with restricted vehicular traffic [37].

The church of *San Domenico Maggiore* is one of the most important religious complexes in Naples (Figure 1a,b). It was built between the XIII and the XIV century by Charles II of Anjou, becoming the motherhouse of the Dominican friars of the Kingdom of Naples and the church of the Aragonese nobility [38]. The church was erected according to the classic tenets of the Gothic style, although this has been compromised due to the numerous interventions that have taken place over the subsequent centuries. These have altered its structure, and the original Gothic forms, through the addition of three naves, side chapels, a large transept and a polygonal apse. The complex has been restored several times; the last restoration started in 2000 and concluded in 2011 [38]. The building is made mainly of Neapolitan yellow tuff, with some other elements, such as the portal, made of marble, the buttresses in *Piperno*, and part of the central space of the apse jutting out covered in red bricks [38].

The cloister of *San Marcellino e Festo* is part of a monastic complex dating back to the early Middle Ages (Figure 1c,d) that has undergone several interventions throughout the centuries [39]. Important restoration work was devised and carried out in 1779 by Luigi Vanvitelli [37,40], particularly affecting the consolidation of the dome, the majolica restoration, the extension of the southern portico of the cloister, and the construction of an oratory. The cloister has a rectangular plan with arches supported by columns made of *Piperno*, with a central monumental garden enriched with fountains and marble sculptures [37]. Since the early twentieth century, the cloister has hosted both the Museum of Paleontology and the Natural Sciences Department of the University of Naples "Federico II". This latter contracted a new restoration intervention in the cloister in 2001.

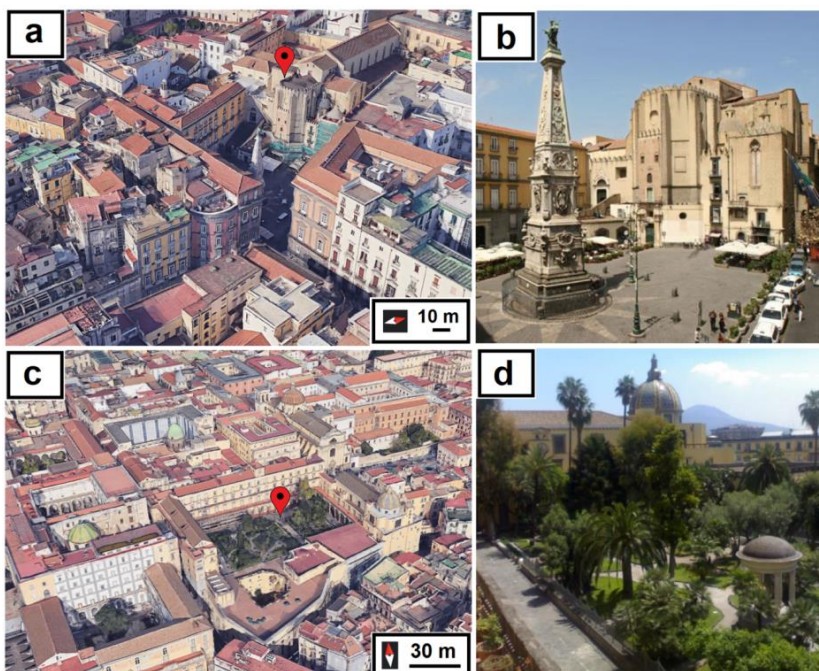

**Figure 1.** Complex of *San Domenico Maggiore* (40°50′55′′N; 14°15′16′′E) and cloister of *San Marcellino e Festo* (Naples, Italy) (40°50′49′′N; 14°15′28′′E): (**a,c**) Aerial view of the two monumental complexes by Google Earth; (**b**) view of the *San Domenico Maggiore* church with the apse facing the eponymous square; (**d**) General view of the *Cloister* and *San Marcellino e Festo* church with evidence of the tiled dome.

The present work focuses on the minero-petrographic and geochemical characterization of black crust samples collected from the above-mentioned sites and investigated using optical and electron microscopy, X-ray powder diffraction and laser ablation inductively coupled plasma-mass spectrometry.

This integrated analytical approach made it possible to obtain the main features of (1) the black crusts and (2) the underlying stone substrate, in terms of micromorphology, mineralogical composition, and major and trace elements. This wide-ranging characterization provided valuable information on the formation processes of black crusts, as well as on interaction between stone substrate and the surrounding environment. Specifically, important environmental data were obtained thanks to the identification of heavy metals, which, as is known, can contribute to the recognition of the main sources of pollution responsible for the deterioration of building materials over time.

## 2. Sampling

Eleven marble fragments were collected from different points at the two study sites; four from the complex of *San Domenico Maggiore*, and seven from the cloister of *San Marcellino e Festo* (Figure 2). Suitable stainless-steel tools, such as lancets and small chisels, were used to sample representative but noninvasive portions of material affected by the presence of black crusts.

With respect to the complex of *San Domenico Maggiore* (Figure 2a–c), samples were retrieved from the façade of the church, specifically from the marble portal located on the south-east side, alongside the apse of the church, overlooking *San Domenico* square.

From first macroscopic observation, the black crusts (SD series) looked rather homogeneous, compact, with a smooth surface and thin thickness. They also showed good adhesion to the underlying stone substrate, which was rather degraded, with evidence of swelling, poor compactness, and sometimes powdery appearance.

With respect to the cloister of *San Marcellino e Festo* (Figure 2d–f), the samples were taken from the large square plan monumental cloister, enriched with a garden adorned with marble fountains, statues, and various artifacts. The seven samples were taken from three different sculptures: one sample from

a marble well (SM-P series) (Figure 2f), three from a marble structure with arches and pillars (SM-A series) (Figure 2e), and three from a female marble bust (SM-S series) (Figure 2d). Additionally, in this case, all black crusts showed a homogeneous and compact morphology, a very thin thickness and good adhesion to the underlying stone substrate. As for the stone substrates, they looked fairly cohesive and slightly altered.

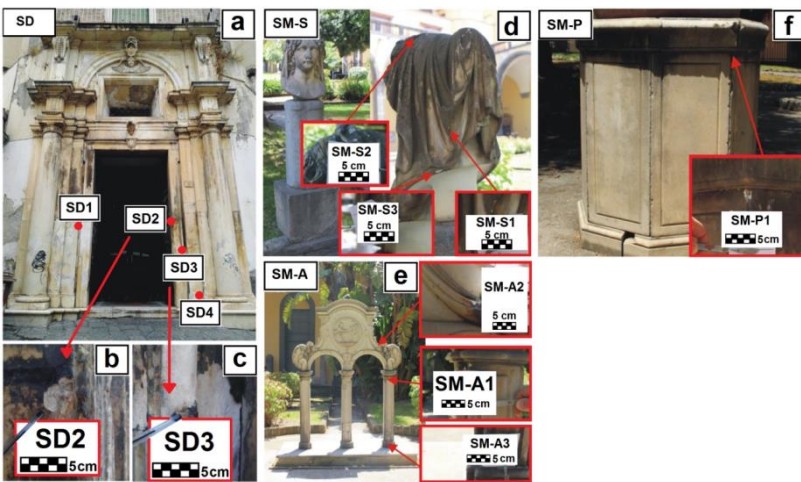

**Figure 2.** Sampling points of fragments collected at: *San Domenico Maggiore* (SD series) (**a**–**c**); Marble bust of the San Marcellino e Festo Cloister (SM-S series) (**d**); Arch sculpture of the *San Marcellino e Festo* Cloister (SM-A series) (**e**); Water well of *San Marcellino e Festo* Cloister (SM-P series) (**f**).

Samples with location description and heights are reported in Table 1.

**Table 1.** List of the examined samples with a brief description. The sampling heights refer to the planking level. The exposition of the surfaces to washout is defined as well exposed (WELL), partially exposed (PART), or not exposed (NOT).

| Sample ID | Sampling Location | Sampling Heights | Exposition to Washout |
|---|---|---|---|
| | **Complex of *San Domenico Maggiore* (SD Series)** | | |
| SD1 | Facade of the San Domenico church, main portal. Sampling on one of the left pillars (looking towards the portal), in a slightly curved area and on a vertical and external surface. | 2.00 m | WELL |
| SD2 | Facade of the San Domenico church, main portal. Sampling on one of the right pillars (looking towards the portal), on a vertical and internal surface. | 2.30 m | WELL |
| SD3 | Facade of the San Domenico church, main portal. Sampling on one of the right pillars (looking towards the portal), on an external corner. | 1.60 m | PART |
| SD4 | Facade of the San Domenico church, main portal. Sampling on one of the right pillar bases (looking towards the portal), on a horizontal surface. | 0.40 m | NOT |
| | **Cloister of *San Marcellino e Festo* (SM-P, SM-A, SM-S Series)** | | |
| SM-P1 | Monumental cloister, well. Sampling on a vertical surface, under the top edge. | 1.00 m | NOT |
| SM-A1 | Monumental cloister, structure with arches and pillars. Sampling on a vertical surface, right column (looking towards the structure). | 1.20 m | PART |
| SM-A2 | Monumental cloister, structure with arches and pillars. Sampling on a convex surface, right side (looking towards the artefact), on a decorative element. | 1.60 m | PART |
| SM-A3 | Monumental cloister, structure with arches and pillars. Sampling on a vertical surface, on the base of the right column. | 0.20 m | WELL |
| SM-S1 | Monumental cloister, female bust sculpture. Sampling on the veil, top of the head, on the back-side. | 1.50 m | PART |
| SM-S2 | Monumental cloister, female bust sculpture. Sampling on the veil, on the head, on the front-side. | 1.50 m | WELL |
| SM-S3 | Monumental cloister, female bust sculpture. Sampling on a vertical portion of the bust, back-side. | 1.50 m | PART |

## 3. Analytical Methods

For a complete characterization of the stone materials and degradation products (i.e., black crusts), several complementary techniques were employed to investigate the textural, morphological and compositional features, as well as the interaction with rock substrate. Optical Microscopy (OM) observations were carried out on polished and stratigraphic thin sections using a Zeiss Axioskop 40 microscope (Carl Zeiss Microscopy GmbH, Jena, Germany, 2007). OM made it possible to: a) determine the textural features; and b) detect weathering rate on superficial layers by evaluating the morphology and growth of black crusts.

Scanning Electron Microscopy (SEM) coupled with energy dispersive X-ray spectrometry (EDS) analyses were performed on polished cross-sections previously covered by carbon coating, to obtain information about micromorphology and chemical composition (in term of major elements) of the black crusts. Analyses were performed with a SEM (Cambridge Instruments, version 360 Stereoscan, UK., Cambridge), equipped with a microanalyzer in energy dispersive spectrometry (EDS) (EDAX model) with an ultrathin (UHT) window in order to ensure the detection of light elements. The operating conditions were set at an accelerating voltage of 20 kV, beam current of 0.2 mA, acquisition time of 100 s, and dead time of 30–35%.

Chemical analyses were carried out according to standard mode and normalized by weight (%). The detection limit for the EDS system is approximately 0.1% weight. The accuracy of the analysis was periodically tested on standard USGS samples. Chemical analyses of the crust surfaces were performed in raster mode.

X-ray powder diffraction (XRPD), performed to investigate the crusts' mineralogical composition, was recorded on an X-ray diffractometer (version D8 Advance, Bruker, UK) using X-ray Cu K$\alpha$ radiation. The operative conditions were 40 kV voltage, 30 mA current, 0.02° 2$\theta$ step size, and 3.0 sec step time with a 2$\theta$ range of 10–80°.

The analysis of trace elements was conducted by using laser ablation–inductively coupled plasma–mass spectrometry (LA-ICP-MS). This method enables the detection and quantification of several elements with spot resolutions of approximately 40–50 µm, leading to the determination of compositional variations on a micrometric scale [41–43]. Measurements were obtained using the LA-ICP-MS instrument (model Elan DRCe, Perkin Elmer/SCIEX, MA, USA), connected to a New Wave UP213 solid-state Nd-YAG laser probe (213 nm). The ablation was performed with spots of 40–50 µm with a constant laser repetition rate of 10 Hz and fluence of ~20 J/cm$^2$. Calibration was performed using the NIST 612-50 ppm glass reference material as the external standard [44]. Internal standardization to correct instrumental instability and drift was achieved using CaO concentrations from SEM-EDS analyses [45]. The accuracy was evaluated on BCR 2G glass reference material and on an in-house pressed-powder cylinder of the standard Argillaceous Limestone SRM1d of NIST [46]. The resulting element concentrations were compared with reference values from the literature [47]. The accuracy, as the relative difference among the reference values, was always better than 12%, and most elements plotted in the range of ±8%. Investigations were performed on cross-sections 100 µm thick. Each sample was subject to several spot analyses, depending on the thickness of the black crust, to assess the compositional variability within the crust and the differences between the degraded portion and the underlying unaltered substrate.

## 4. Results

### 4.1. Optical Microscopy (OM) and Mineralogical Analysis (XRD) of the Stone Materials and the Black Crusts

Characterization was performed by optical microscopy (OM) of the samples collected at the complex of *San Domenico Maggiore* (SD series) and the cloister of *San Marcellino e Festo* (SM-P, SM-A, SM-S series). Observations, performed on the most representative samples (i.e., SD2, SD3, SM-P1, SM-A1 and SM-S1) are reported in Table 2, and the samples were classified on the basis of their textural features along with mineralogical substrate composition.

With respect to the stone substrate, all samples could be classified as marbles with fine grain size (Maximum Grain Size < 1 mm) (Figure 3); the fabric could be defined as "mosaic" type, with tiny to very tiny crystal size, often forming triple junctions at 120° [48]. The texture was homeoblastic (Ho) in the SD2 and SD3 (i.e., samples from the complex of *San Domenico Maggiore*—portal) and SM-P1 (i.e., sample from the cloister of *San Marcellino e Festo*—well) samples, while it was heteroblastic (He) in SM-A1 and SM-S1 (i.e., samples from the cloister of *San Marcellino e Festo*—structure with arches and pillars and the female bust, respectively). The grain size ranged from 0.1 mm to 0.45 mm in SD2 and SD3, from 0.1 mm to 70 mm in SM-S1, from 0.1 mm to 0.47 mm in SM-A1, and from 0.1 mm to 0.55 mm in SM-P1. All samples also showed remarkable micro-cracks just at the substrate–crust interface; the progress of this fracturing process will most probably result in a complete detachment of the weathered portion.

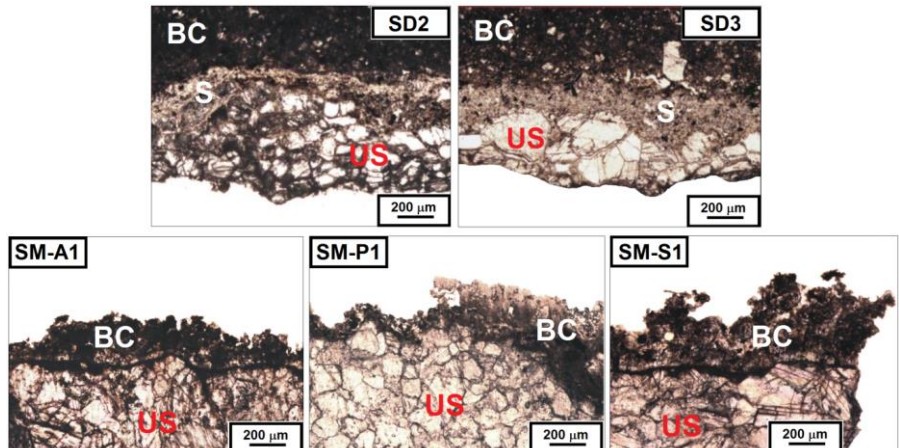

**Figure 3.** Microphotograph (OM, PPL) showing textural features of some selected marble fragments, with evidence of the black crust layer on the surface. SD2, SD3 are samples from the complex of *San Domenico Maggiore*, while SM-A1, SM-P1 and SM-S1 were taken from the cloister of *San Marcellino e Festo*. Notes: unaltered substrate = US; black crust = BC; scialbo = S.

As for the black crusts, samples SD2 and SD3 (i.e., samples from the complex of *San Domenico Maggiore*—portal) showed three different layers (Figure 3, samples SD2, SD3). Starting from the most external: a) a first layer of black crust with a homogeneous morphology, thickness ranging between approximately 350 and 10 μm, and a color (PPL, plane polarized light) varying from light brown to dark brown. Embedded iron oxides and hydroxides were observed, together with black combustion particles (i.e., particles formed during combustion processes, containing sulphur compound and catalysts; once wet, such particles will nucleate gypsum crystals and will remain embedded in a gypsum crust) of spherical, sub-spherical and prismatic shape (ranging in size from 80 to 10 μm); b) a layer of *scialbo* which shows a good adherence to the substrate, with relatively homogeneous morphology, and a thickness ranging from 3 mm to 800 μm, dark brown in color, a cryptocrystalline aspect, a very fine granulometry and a secondary porosity of about 10% related to the dissolution of some portions. Otherwise, only rare individual and small calcite crystals could be identified in addition to the oxides; c) another layer of crust (Figure 3) showing good adherence to the stone substrate and a compact morphology, thin thickness varying from about 400 to 60 μm, and a light grey color. The latter layer displays microcrystalline gypsum crystals, iron oxides and hydroxides, and spherical, sub-spherical and prismatic black combustion particles (ranging in size from 80 to 10 μm) inside.

With respect to the samples collected from the cloister of *San Marcellino e Festo*, the SM-S1 and SM-A1 crusts had an irregular morphology with a jagged outer edge, mammelonar in some portions, with thickness ranging from 520 to 75 μm. The color varied from dark grey to brown, due to the presence of iron oxides and hydroxides respectively. Additionally, spherical, sub-spherical and

prismatic combustion particles (ranging in size from 85 to 10 μm) distributed evenly along the entire investigated surface were visible (Figure 3 sample SM-S1).

Instead, SM-P1 crust (Figure 3) shows a heterogeneous morphology, mostly compact, with a thickness ranging from 800 and 50 μm. Inside, iron oxides, hydroxides and well-distributed combustion particles (sizes between 125 and 25 μm) were recognized, providing an overall dark grey color.

With respect to mineralogical composition, these results are reported in Table 2. XRPD analyses were performed separately on the substrate and on the crusts, evidencing that calcite is the main mineralogical phase in the substrate (marble), whereas gypsum, whewellite and calcite traces were present in all of the crusts. Some differences were highlighted for samples SD3 and SM-A1 (presence of quartz and weddellite) along with SM-S1 and SM-P1 (presence of weddellite).

According to the literature [49], the presence of calcium oxalates, i.e., whewellite and weddellite, may be a result of the restoration work carried out on the artifacts in the past [50], or could be linked to biological activities or other natural reactions [51].

**Table 2.** Main textural features (OM) and mineralogical phases (XRPD) occurring in the analyzed samples, considering both unaltered substrate and black crusts.

| Complex of *San Domenico Maggiore* (SD series) | | | | | | | | | |
|---|---|---|---|---|---|---|---|---|---|
| Sample ID | Grain Size μm | Texture | Fabric | Mineralogical Phases in the Substrate | Mineralogical phases in the black crust | | | | |
| | | | | | Cal | Gp | Qz | Whw | Wed |
| SD2 | 450–100 | Ho | Mosaic | Cal | +++ | ++ | − | + | − |
| SD3 | 450–100 | Ho | Mosaic | Cal | +++ | ++ | + | ++ | − |
| Cloister of San *Marcellino e Festo* (SM-P, SM-A, SM-S series) | | | | | | | | | |
| Sample ID | Grain Size μm | Texture | Fabric | Mineralogical phases in the substrate | Mineralogical phases in the black crust | | | | |
| | | | | | Cal | Gp | Qz | Whw | Wed |
| SM-P1 | 550–100 | Ho | Mosaic | Cal | ++++ | +++ | − | ++ | + |
| SM-S1 | 690–100 | He | Mosaic | Cal | ++++ | +++ | − | ++ | + |
| SM-A1 | 470–100 | He | Mosaic | Cal | ++++ | ++ | + | + | + |

Minerals abbreviations according to [52]: Cal. calcite; Qz. quartz; Gp. Gypsum; Whw. Whewellite; Wed. Weddellite; ++++. very abundant; +++. abundant; ++. moderate; +. poor; -. not present.

### 4.2. Micromorphological and Elemental Analysis of the Black Crusts by SEM-EDS

Scanning Electron Microscopy observations highlighted many similarities between SD samples. Both show a stone substrate (US unaltered) overlayed by well adherent and homogeneous crusts with different thicknesses (Figure 4a,b). The SD2 crust has a size that varies from 175 to 10 μm (Figure 4a), while SD3 ranges from 150 to 5 μm (Figure 4b).

Black crusts consist of gypsum, which has an acicular and lamellar crystal habit, in which combustion particles of various sizes and morphologies were identified (Figure 4c,d). Particles have diameters ranging from 80 to 3 μm, and they are spherical, sub-spherical, and irregular in shape, displaying smooth, porous, or rough surface (Figure 4c). These are homogeneously distributed over the whole examined surface.

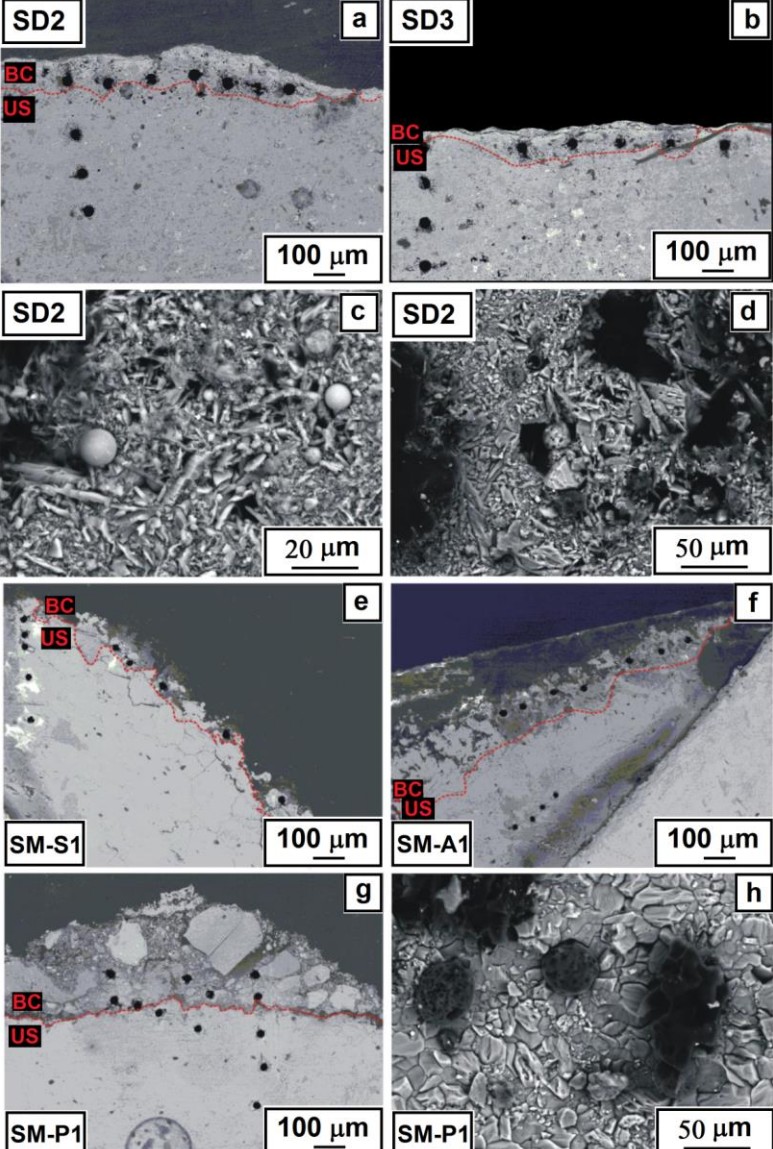

**Figure 4.** SEM microphotographs of some examined samples (i.e., SD2, SD3, SM-S1, SM-A1 and SM-P1) with details of combustion particles and gypsum crystals (**c,d,h**). The holes caused by LA-ICP-MS spot analyses (carried out on black crusts = BC and unaltered substrates = US) are also visible (**a,b,e,f,g**). The red dotted line demarcates the layer of black crust from the substrate. Elemental analyses were carried out by EDS on the black crust to evaluate the chemical composition in terms of major elements, as well as their distribution within the sample (i.e., black crust).

Micromorphological investigations on SM-S1 (Figure 4e) showed a crust with variable thickness (from 520 to 75 μm), irregular and heterogeneous morphology. Inside, acicular-lamellar gypsum crystals and numerous combustion particles were recognized; the latter are characterized by a morphology ranging from sub-spherical microporous to smooth spherical. The crust appears to be well adherent to the underlying substrate, which is degraded and characterized by many micro-fractures.

The crust in SM-A1 (Figure 4f) is well adherent to the substrate, with higher thickness than the other samples (up to 350 μm), a jagged outer edge, and a heterogeneous morphology. Acicular and lamellar gypsum crystals and few combustion particles were also recognized. These particles, ranging in size from 70 to 5 μm both have partly sub-spherical morphology and porous surface and partly irregular morphology and rough surface.

Finally, SM-P1 (Figure 4g,h) displays a poorly degraded stone substrate, with a well adherent crust only at some points of the analyzed surface. The crust shows a heterogeneous and irregular morphology, with a thickness ranging from 790 to 50 μm (Figure 4g), where lamellar gypsum and calcite crystals could be identified, probably coming from the stone substrate. Additionally, it combustion particles with different shapes (spherical, sub-spherical and irregular), sizes (thickness between 2.5 and 125 μm) and surface morphologies (smooth, porous and wrinkled) can be recognized, distributed variously along the entire surface investigated (Figure 4h).

Black crusts are mainly composed of $SO_3$ and CaO, clearly attributable to the gypsum. Moreover, $SiO_2$, $Al_2O_3$, $Fe_2O_3$ were detected, along with smaller amounts of $K_2O$, $P_2O_5$, $Na_2O$, MgO and $TiO_2$, whose presence is ascribable to the abovementioned particles being embedded in the crust (Table 3).

**Table 3.** Average concentrations of major elements expressed in oxides (wt%) in the black crust of sample SD2 as representative of the complex of *San Domenico Maggiore* and SM-P1 as representative of the cloister of *San Marcellino e Festo.* Measurements were obtained by SEM-EDS analysis performed in raster mode.

| | Complex of *San Domenico Maggiore* | Cloister of *San Marcellino e Festo* |
|---|---|---|
| **Major Elements** | **Sample SD2** **Average Analysis No. 8** | **Sample SM-P1** **Average Analysis No. 8** |
| **Na₂O** | <0.1 | 1.99 |
| **MgO** | <0.1 | 1.19 |
| **Al₂O₃** | 6.70 | 8.82 |
| **SiO₂** | 7.82 | 13.60 |
| **P₂O₅** | 0.99 | <0.1 |
| **SO₃** | **40.90** | **38.60** |
| **K₂O** | 3.49 | 2.50 |
| **CaO** | **35.50** | **28.40** |
| **TiO₂** | <0.1 | 0.57 |
| **Fe₂O₃** | 5.59 | 4.36 |

Since the crusts in the SD2 and SD3 samples show a similar composition, as do SM-S1, SM-P1 and SM-A, Table 3 only reports the crust compositions of the SD2 and SM-P1 samples, as representatives.

The distributions of $SO_3$ and CaO, detected in the analyzed window for SD2 and SM-P1 were obtained by means of false color maps regarding S and Ca (Figure 5a–c), as well as their combination (Figure 5d).

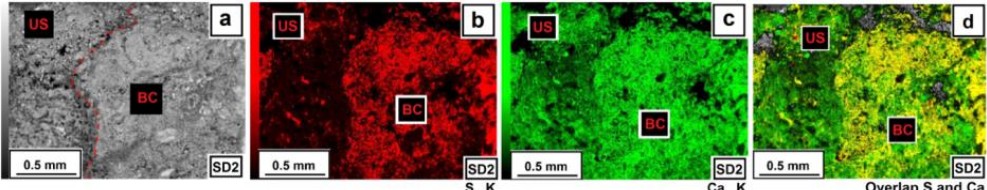

**Figure 5.** False color maps by SEM-EDS related to SD2 sample as representative. (**a**) EDS analyzed window of the SD2 sample, consisting of the crust (BC) and stone substrate (US); (**b**) distribution of S (red); (**c**) distribution of Ca (green); (**d**) overlapping and distribution map of S and Ca.

Figure 5d further confirms that the BC surface is mainly composed of gypsum, with calcium and sulfur being the main components, with a more marked yellow coloring resulting from the combination of red and green colors (i.e., overlapping of red colors indicating 100% S and green indicating 100% Ca, according to their greater or lesser amount). Additionally, the US surface shows a high concentration of calcium oxide, demonstrated by the greener coloring.

### 4.3. Trace Elements Analysis by LA-ICP-MS

Chemical characterization of samples in terms of trace elements was performed using the LA-ICP-MS technique on cross-sections with polished surface.

This method made it possible to determine the trace elements in both the black crusts and the unaltered substrate (i.e., marbles), selecting a statistically valid number of analyses as a function of the sample thickness. Exceptionally, the assays were only carried out on the crust in SM-P1, due to the lack of representativeness of the substrate resulting from the insufficient thickness available.

The average concentrations (in ppm) of the most representative chemical elements, both in the unaltered substrates (US) and in the black crusts (BCs), is reported in Table 4.

Chemical elements such as Ba, Cu, Pb, Ti and Zn achieved the major concentrations in the black crusts (Table 4).

Specifically, the BCs (SD2 and SD3) taken from the complex of *San Domenico Maggiore* show higher average maximum values in Pb (3525 ppm), Zn (1580 ppm) and Cu (224 ppm) than the substrate, where they are considerably lower (i.e., Pb is 82.09 ppm; Zn is 32 ppm; Cu is 18.18 ppm). In the same samples of BCs, similar and sometimes lower values of Ba, As and V were detected with respect to the unaltered substrate (Table 4).

The BCs from the cloister of *San Marcellino e Festo* (SM-S1, SM-A1 and SM-P1) display lower concentrations in heavy metals compared to the previous ones (i.e., SD2 and SD3). In detail, BCs of SM-S1, SM-A1 and SM-P1 show higher average maximum values in Ba, Ti, Zn and Cu than the unaltered substrate (Table 4). Specifically, in SM-A1, Ba increases from 9.15 ppm in the US to 288 ppm in the BC; in SM-S1, Ti varies from 12.38 ppm in US to 417 ppm in BC, Zn from 19.25 ppm in US to 321 ppm in BC, Cu from 4.4 ppm in US to 93.55 ppm in BC (Table 4).

The chemical compositions determined for the samples from the complex of *San Domenico Maggiore* suggest a certain correlation between the concentrations of the trace elements and the position of the samples in terms of height and exposure to washout (Figure 2 and Table 1). In particular, the trace elements detected in SD2, taken at 2.30 m high, show greater overall concentrations than SD3, sampled at 1.60 m high (Figure 2). In fact, this latter one is located in an area only partially exposed to washout (i.e., on a vertical and internal surface), thus favoring a major accumulation of pollutants over time.

Conversely, the black crusts belonging to the three historical artifacts from the cloister of *San Marcellino e Festo* (SM-S1, SM-A1 and SM-P1) do not show a clear correlation between the concentration of the trace elements and their sampling position.

Overall, the samples of the cloister exhibit lower concentrations than those of *San Domenico Maggiore*, which is attributable both to different sampling heights and to exposure conditions. In fact, the artifacts in *San Marcellino e Festo* are located in a more protected micro-environment with respect to the samples of *San Domenico Maggiore*, which are directly exposed in a higher vehicular traffic area.

**Table 4.** The average concentration resulted from five point-analysis (expressed in ppm) of the most representative chemical elements, both in the unaltered substrate (US) and in the black crusts (BC).

| Element | BC-SD2 | | BC-SD3 | | US-SD | | BC-SMS1 | | US-SMS1 | | BC-SMA1 | | US-SMA1 | | BC-SMP1 | | US-SMP1 | |
|---|---|---|---|---|---|---|---|---|---|---|---|---|---|---|---|---|---|---|
| | Amount (ppm) | Std Dev | Amount (ppm) | Std Dev | Amount (ppm) | Std Dev | Amount (ppm) | Std Dev | Amount (ppm) | Std Dev | Amount (ppm) | Std Dev | Amount (ppm) | Std Dev | Amount (ppm) | Std Dev | Amount (ppm) | Std Dev |
| As | 48.46 | 2.10 | 36.52 | 2.17 | 51.63 | 12.23 | 5.59 | 0.60 | 1.72 | 0.51 | 7.89 | 1.36 | 2.33 | - | 9.09 | 1.46 | 2.70 | 0.73 |
| Ba | 938 | 91.18 | 1106 | 73.74 | 1104 | 232 | 139 | 32.44 | 6.93 | 1.68 | 288 | 26.09 | 9.15 | 0.30 | 190 | 67.82 | 16.27 | 7.71 |
| Cd | 1.19 | 0.07 | 0.86 | 0.07 | 0.54 | 0.10 | 0.59 | 0.07 | 0.77 | 0.16 | 0.58 | 0.17 | 0.44 | - | 1.44 | 0.15 | 0.63 | 0.22 |
| Cr | 22.75 | 1.14 | 17.80 | 0.69 | 26.94 | 6.24 | 12.00 | 1.97 | 5.22 | 0.71 | 20.89 | 0.91 | 15.31 | - | 8.71 | 1.17 | 2.89 | 0.61 |
| Cu | 224 | 24.61 | 60.08 | 4.02 | 18.18 | 3.69 | 93.54 | 21.10 | 4.44 | 2.69 | 26.17 | 6.23 | 1.23 | 0.28 | 45.06 | 8.40 | 52.32 | 5.97 |
| Ni | 10.31 | 0.61 | 8.02 | 0.77 | 7.68 | 1.45 | 11.54 | 2.73 | 1.06 | 0.29 | 3.56 | 0.65 | 0.68 | 0.20 | 6.06 | 1.18 | 10.44 | 0.90 |
| Pb | 3525 | 262 | 3134 | 292 | 82.09 | 23.49 | 85.55 | 7.87 | 17.37 | 8.04 | 109 | 6.67 | 37.02 | 5.20 | 122 | 11.69 | 147 | 6.36 |
| Sb | 25.59 | 1.75 | 12.71 | 1.21 | 6.73 | 1.15 | 5.84 | 1.07 | 0.19 | 0.01 | 1.65 | 0.41 | 0.11 | 0.04 | 2.01 | 0.94 | 0.48 | 0.24 |
| Ti | 1377 | 75.12 | 1023 | 63.80 | 1842 | 281 | 417 | 62.09 | 12.38 | 2.96 | 110 | 35.41 | 20.96 | 12.52 | 492 | 134 | 53.15 | 26.08 |
| V | 37.29 | 2.13 | 31.75 | 4.16 | 37.11 | 4.95 | 16.67 | 1.68 | 0.96 | 0.16 | 10.65 | 0.96 | 0.78 | 0.27 | 18.23 | 3.81 | 5.66 | 3.18 |
| Zn | 1580 | 197 | 589 | 52.97 | 32.00 | 2.07 | 321 | 26.73 | 19.25 | 7.05 | 169 | 53.64 | 26.62 | 1.74 | 481 | 62.21 | 904 | 24.13 |

## 5. Discussion

The enrichment factors (EFs) of chemical elements were calculated as the ratios of crust/substrate metals' concentration in order to discriminate the origin of the elements respectively attributable to the deposition process, or to the substrate. The results are reported in Table 5.

**Table 5.** Enrichment factors (EFs) of the chemical elements detected in the black crusts: EFs > 1 (values in green cells); EFs = 1 (values in white cells); EFs < 1 (values in red cells).

| Chemical Elements | Complex of *San Domenico Maggiore* | | Cloister of *San Marcellino e Festo* | | |
|---|---|---|---|---|---|
| | Sample SD2 | Sample SD3 | Sample SM-S1 | Sample SM-A1 | Sample SM-P1 |
| As | 0.9 | 0.7 | 3.3 | 4.6 | 5.2 |
| Ba | 0.8 | 1 | 20.1 | 41.6 | 38.4 |
| Cd | 2.2 | 1.6 | 0.8 | 0.8 | 5.5 |
| Co | 2 | 0.9 | 12.4 | 3.9 | 11.4 |
| Cr | 0.8 | 0.7 | 2.3 | 4 | 1.2 |
| Cu | 12.3 | 3.3 | 21.1 | 5.9 | 9.5 |
| Ni | 1.3 | 1 | 10.8 | 3.4 | 6.4 |
| Pb | 42.9 | 38.2 | 4.9 | 6.3 | 5.6 |
| Sb | 3.8 | 1.9 | 30.7 | 8.7 | 10.6 |
| Ti | 0.7 | 0.6 | 33.6 | 8.9 | 22.5 |
| V | 1 | 0.9 | 17.4 | 11.1 | 9.6 |
| Zn | 49.4 | 18.4 | 16.6 | 8.8 | 16.8 |

Generally, an enrichment factor lower than or equal to 1 suggests that the source of the element comes from the substrate, while an enrichment factor greater than 1 may be imputable to an external "provenance", meaning that elements are linked to a deposition process. This secondary provenance is more accentuated as the enrichment factor increases. Considering Table 5, it is possible to observe these differences; specifically, cells in green show EFs values greater than 1, while red ones refer to values lower than 1.

The high amount of Ba in the samples (Table 5) from the cloister of *San Marcellino e Festo* (SM-P, SM-A, SM-S series) is probably a result of restoration interventions [53] carried out in the past, as barium hydroxide was widely used for the conservative treatment of marbles in the past [54].

This thesis is supported by the investigations carried out using OM. Thin section observations made it possible to highlight the presence of a layer of *scialbo* between the marble substrate and the black crust (i.e., Figure 3, sample SD3). This could be attributed to a restoration intervention, since this practice is rather common in restoration of marble. The enrichment in barium is further provided by EF values (Table 5).

In particular, different EFs can be ascribable to various factors such as the type of stone material, the particle deposit morphology (vertical surfaces), the different exposure to emission sources (mobile or fixed) and the wash-out phenomenon. In this way, specimens SD2 and SD3 showed a high enrichment factor for some elements such as Pb, Zn and Cu, because they were sampled from the church portal, which has been affected by mobile polluting sources—specifically, vehicles—in the past [55].

Figure 6, with respect to the complex of *San Domenico Maggiore* (Figure 6a–c), shows good correlations among the different heavy metals, such as Zn vs. Pb, Cu vs. Zn and Sb vs. Cu. In detail, the high correlation between Zn vs. Pb ($R^2$ = 0.8022) is in agreement with the use of leaded gasoline as fuel for auto-motion [4], used up until a few decades ago. Moreover, the correlation between Cu vs. Zn ($R^2$ = 0.8406) suggests an enrichment of the samples with tire wear particles and other artefacts of friction and machine wear [56–59]. Additionally, the latter could indicate soil contamination from lubricating oil and exhaust emissions from vehicles [60,61]. Finally, the good correlation between Sb vs. Cu ($R^2$ = 0.884) indicates a contribution from the brake wear [62,63].

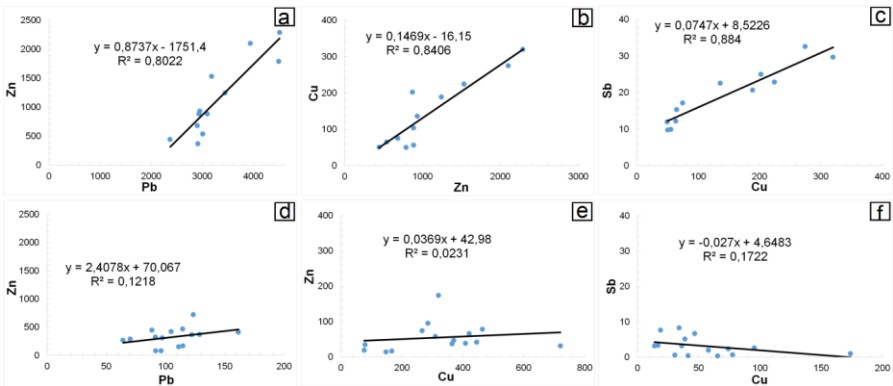

**Figure 6.** Correlation between the Zn vs. Pb, Cu vs. Zn e Sb vs. Cu of the two different sites: (**a**–**c**) Complex of *San Domenico Maggiore*; (**d**–**f**) Cloister of *San Marcellino e Festo*.

On the contrary, the samples from the cloister of *San Marcellino e Festo* (SM-S1, SM-A1 and SM-P1) show a low correlation between the heavy metals (Figure 6d–f), suggesting differences in the source.

As, Cr, Cu, Ni, Pb, Sb, Ti, V and Zn registered a moderate enrichment in all black crust samples (3 < EF < 33.6), which could be associated with different anthropogenic sources such as industrial activities [64–66], pavement wear [67], or vehicular traffic.

Previous studies carried out on atmospheric particulate matter and dust sampled from the soil in the Naples area [68,69] highlighted the presence of two different pollutant sources: mobile and fixed. In particular, the latter one is related to the ILVA steel mill (working until 1993) located in the industrial area of Bagnoli-Coroglio and to the oil refineries (i.e., Q8, Esso, Tamoil) in the Eastern area of Naples. These industrial emissions have certainly affected enrichment in some elements such as Ti, Zn, Pb, Cu, Ni and V. Furthermore, the presence of high Sb values in these samples could be attributed to the incinerator plants [70] located 30 km north of the city [71,72].

Additionally, by comparing the lower contents of As, Ni and V determined in this study with respect to previous research [23,26,27,30,32,33,73], it is possible to assert that these values may be dependent on the moderate degree of use of domestic heating resulting from the more temperate climate of Naples compared to that of other Italian and European cities.

In fact, the above-mentioned elements may be generally associated with the different fuels used in heating, such as coal (in the past) or fuel oils [74,75].

Considering the data obtained, it is possible to assert that the greatest enrichment in heavy metals of the black crusts from the church of *San Domenico Maggiore* (samples SD2 and SD3) is certainly due to the use of Pb-based gasoline, as well as tire and brake wear and other artefacts of machine wear. This aspect is also evident in Figure 7, which shows the location of the two case studies in the center of Naples. In fact, although the church of *San Domenico Maggiore* is nowadays located in a pedestrian area (i.e., *Piazza San Domenico Maggiore* and *Vico San Domenico Maggiore*), it was surrounded by a high volume of vehicular traffic until a few decades ago.

With respect to the samples from the cloister of *San Marcellino e Festo* (SM-S1, SM-A1 and SM-P1), data show an enrichment in various heavy metals (As, Cr, Cu, Ni, Pb, Sb, Ti, V and Zn) that can be traced to different polluting sources (industrial activities and vehicular traffic). The sampled artifacts are located inside a closed cloister and not exposed to direct sources of emissions, but they still showed weathering phenomena related to stone decay and air pollution.

The results obtained indicate that all monuments, even those not directly exposed to sources of direct emissions, can undergo decay processes. This demonstrates how the conservation of built cultural heritage is strictly connected to environmental protection issues and how it is necessary to intensify the related intervention policies.

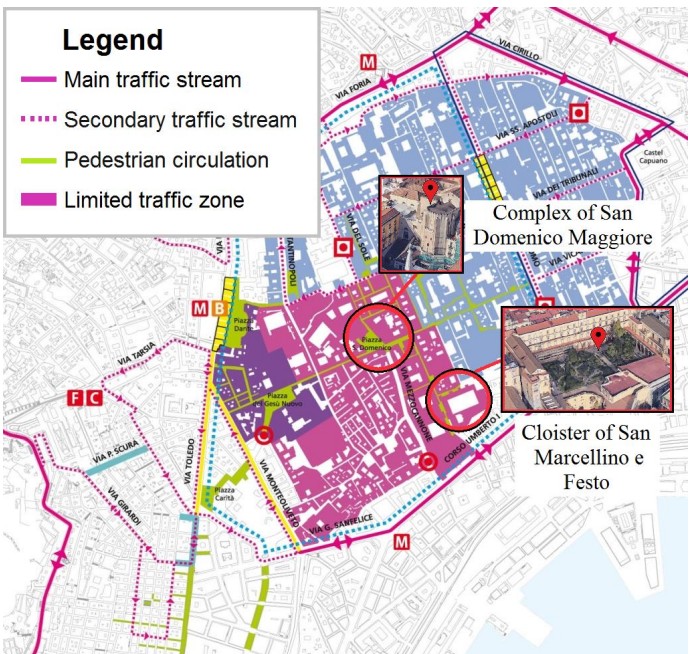

**Figure 7.** Road map of the urban city center of Naples. The red circle indicates the location of the sampled monuments: SD = Complex of *San Domenico Maggiore*, SM = Cloister of *San Marcellino e Festo*.

## 6. Final Remarks

Black crusts collected from two historical sites with different exposure located in the center of Naples were analyzed in order to detect the variability in the degradation forms, mainly due to atmospheric pollutants.

The minero-petrographic and geochemical investigations made it possible to:

- determine that the concentration of specific elements (such as As, Sb, Pb, Zn, Cu, Sn, etc.) was noticeably higher in the samples coming from the complex of San Domenico Maggiore, evidencing the fingerprint of air pollution due to vehicular emissions.
- show that the As amount detected in Naples city center was lower than in other Italian and European cities studied in previous research, highlighting the importance of the impact of the local pollution sources on the cultural heritage.
- consider the evolution of the conservation state of the rock substrate. Some elements such as Zn, Cu, Ni, etc. were more abundant in the substrate, evidencing the presence of a network of microcracks favoring the migration of chemical elements from the crusts to the substrate. This mobility can also lead to the formation of new crusts, contributing to the acceleration of weathering damage.

The results obtained represent a further milestone for better managing future restoration interventions, especially in terms of the choice of the best cleaning procedures for historical and monumental complexes. Additionally, suitable consolidation procedures will make it possible to increase the resistance of stone materials against the degradation phenomena mainly related to the geochemical mobility from the black crusts to the substrate.

**Author Contributions:** Conceptualization, M.F.L.R.; P.C. and S.F.G; methodology, D.B.; M.F.L.R.; formal analysis, M.R.; V.C.; S.A.R.; N.R. and L.R.; investigation, M.R.; V.C.; S.A.R.; L.R.; C.R.; C.G.; data curation, M.R.; V.C.; writing—original draft preparation, M.R. and V.C.; writing—review and editing, M.F.L.R.; P.C.; S.F.G. and N.R.; supervision, M.F.L.R.; P.C. and S.F.G.; funding acquisition, M.F.L.R. All authors have read and agreed to the published version of the manuscript.

**Conflicts of Interest:** The authors declare no conflict of interest.

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
