# Peer review of "Multidisciplinary Approach for Evaluating the Geochemical Degradation of Building Stone Related to Pollution Sources in the Historical Center of Naples (Italy)"

_applsci, doi:10.3390/app10124241_

Round 1

Reviewer 1 Report

This is an interesting manuscript that deals with the causes of formation of black crusts on two historical buildings of Naples (Italy). The authors carry out their research focused on the causes and formation processes of these black crust comparing the two monuments exposed to different air pollution sources, namely in terms of car traffic pollution. The study of formation of black crust has been extensive and the process of formation is well-known, being most of the problematic questions focused on the pollution source (traffic, industrial, domestic heating, …). This is a difficult task difficult to demonstrate due to confounding factors and that some air pollution sources have similar composition. So, the approach of the authors is correct. From a formal point of view analytical techniques used and execution of the work is impeccable. The authors have taken some samples from both buildings, studied some of them with the same methods, and compared results. Them, they discuss results with references and provide conclusions. However, this part of the work must be improved. In the present format, the authors join the results and discussion in a single section. In this subject, it results particularly important to separate both, due to the extensive knowledge that exist. So, the manuscript can be accepted for publication after major revision, focused in this part of the work.

I include some major and minor comments to improve the work.

Major comments:

From my point of view the most necessary change is to separate the results from the discussion. The study of formation of black crust has been extensive and the most problematic questions focus on the pollution source, voiding confounding factors. On one hand, this is relatively easy: the authors provide their results from the beginning of section 4 (line 184) to page 11 line 333. I think that this body of text is clearly the text that describes results and does not require major improvements.

On the other hand, the rest of the section would be the discussion but this part should be improved and extended incorporating some of the text of the section ’Final Remarks’. The final remarks should be shortened. The text from lines 426 to 438 should be incorporated to the discussion. In this part, I see the most weak point of the work (from my point of view). The authors refer to previous restorations that result in the presence of some element in the black crusts. This is probably because they have some information of such restoration works. This is one of the confounding factors that should be clarified and more data should be provided here. When such restoration were performed in terms of time? There are information on the treatments used and chemicals added to the rock? An interesting question that is not done is that if the authors know the time of the former restoration they could assess a ‘decay rate’ caused by air pollution. An average or range of thickness for the crust would even assessed… is this possible? This kind of information will be completely new and interesting!

Minor comments:

Abstract: from line 24 to 27 the information provided is not significative but common information. Please, change or remove.

Line 64: the authors wright “In this paper, the analysis of black crusts coming from two monumental and historical sites located in the city of Naples is reported.” but in Line 76 the authors wright “The choice of the two sites with different exposure in the Naples urban context was made to detect variability in the degradation forms, mainly due to the air pollution phenomena.” However, the authors just focus the work on black crust. Please, correct.

Figure 2. I think this figure is very important for the work but in the present format it is difficult to observe the locations and details. Could you provide a bigger picture? A A4-sized figure could be more useful.

Figure 3. In the first line of pictures (SD2 and SD3) the the photos are inverted with respect to the others. Please, provide the same orientation for all. It could be useful if the authors could mark the stone and BC layers.

Line 237: calcium oxalates have also been linked to biological growths (there are several references) and other causes (10.1179/sic.1991.36.1.24). This can be a confounding factor. Please, include this point.

Lines 317-320. The authors provide the amounts of some elements in BCs obtained from analysis, but this amounts are difficult to assess in bulk terms. It could be better if they provide the amounts of the same elements for the substrate.

Table 5. Please remark the meaning of the green and red colours in the table caption.

Lines 371-378: An interesting reference on air pollution and monuments refer to element  composition of fuels and other sources (10.1016/j.conbuildmat.2010.07.001). It also provide some other references therein for other sources (domestic heatings, brakes, tires,…). Please, check.

Lines 382-384: the industrical activities can provide a very wide range of pollutants. It could be important to provide more particular data on the particular industial activities that could affect to the studied buildings. Some other questions should be considered here and in lines 392-395, such as the distance of buildings to industrial sources (is it the same?), the main winds, … Al of them are crucial to attribute this kind of pollution to the decay in the studied buildings. If not it results speculative.

Author Response

all revisions are in the attached file

Reviewer 2 Report

This is an interesting paper putting in relation pollution phenomena with black crusts growth. The discussion regarding the results coming from LA-ICP-MS constitutes the original part. The method described is not new, but it could be of interest because never applied to the historical centre of Naples.

Some minor revisions are suggested in the attached file. 

Please limit the self citations

Author Response

all revisions are in the attached file

Reviewer 3 Report

The present research reports the characterization of black crusts on ancient monuments in the city of Naples (Italy).

This topic has been extensively investigated in the past and therefore a huge bibliography on this weathering phenomenon is available. In particular, black crusts on marbles have been extensively investigated in the past by means of laboratory and innovative analytical techniques.

 In the introduction the Authors report that, for the first time, this weathering phenomenon has been characterized in the city of Naples but, in the discussion, a comparison between the black crusts observed in other Italian cities is lacking. Such comparison must be reported and discussed in details.

Two further main aspects of weakness of this article can be found in the way the experimental part has been conducted. The former is related to the XRPD results where the phase contents have been extrapolated by means of the peaks intensity only. At present there are lots of user-friendly and free software that allow to extract the phase contents from the XRPD patterns. In fact, in these types of materials calcite and gypsum show preferred orientation which largely affects the accuracy of the quantitative XRPD results. The latter refers to the chemical results. The actual relationship between the contents of the minor/trace elements would gain in accuracy if some statistical analysis (i.e. multicomponent analysis) would have been performed.

Furthermore, the text presents both many repetitions, as well as the use of the language is not fluid and would require an extended review.

In addition to the above general observation, I have highlighted directly on the manuscript, which you will find as attached file, the parts that, in my view, are not clear.

 In conclusion, I regret, but in the present form I cannot recommend this paper for publication. The text does not exhibit a character of novelty without a comparison with the existing literature data, essential to support the discussion.

Author Response

all revisions are in the attached file

Round 2

Reviewer 3 Report

I noticed that some corrections have been done and the paper has improved. If the other reviewers accept the new version, I have no problems.